# New Technology and Experimental Research on Thick-Walled Tube Fatigue Impact Loading Precision Separation

**Ren-Feng Zhao \*, Wei-Cheng Gao, Dong-Ya Zhang, Xu-Dong Xiao, Yan-Wei Liu and Run-Ze Pan**

School of Mechanical and Precision Instrument Engineering, Xi'an University of Technology, Xi'an 710048, China; gaoweicheng1997@hotmail.com (W.-C.G.); dyzhang@xaut.edu.cn (D.-Y.Z.); xiaoxd@xaut.edu.cn (X.-D.X.); liuyw@xaut.edu.cn (Y.-W.L.); p754004757@hotmail.com (R.-Z.P.)
\* Correspondence: zrf20070607@xaut.edu.cn

**Abstract:** Traditional separation methods for thick-walled metal tubes include turning and sawing, which suffer from wasted raw material and low efficiency. In view of this, this paper proposes a new process of using impact load to promote crack generation and tube separation. Based on the principles of radial repeated impact load, stress concentration effect and fatigue fracture, the rapid initiation and stable expansion of tube fatigue crack are promoted. In addition, the crack initiation mechanism of the tube V-notch root cracks under radial repeated load when the tube is in a restrained state. For the experimental study of the GCr15 steel tube, a multistep decline frequency time tube separation control curve with an initial frequency from 4 Hz to 31 Hz and termination frequency from 1 Hz to 8.5 Hz was designed, and the precision tube separation device is loaded by pneumatic fatigue shock to achieve tube precision separation. In addition, a tube fracture quality evaluation method is proposed. According to the test results, the stress concentration effect of V-notch can significantly reduce the average stress in the process of tube fatigue separation and accelerate the generation of microcracks. Under the continuous action of repeated impact load, the loading method of multistep decline can effectively control the rapid crack initiation and stable expansion of the GCr15 tube V-notch root crack. Moreover, the tube final fracture region has relatively small defects, which can obtain good fracture quality.

**Keywords:** thick-walled tube; repeated impact load; stress concentration effect; crack initiation; control curve

## 1. Introduction

With the rapid development of high-tech fields such as aviation, aerospace, advanced rail transit and automobiles, there is an urgent need for efficient and precise separation technology of metal tubes of different materials and sizes. The blanking of metal tubes is one of the initial processes in the manufacturing process of a large number of mechanical parts. The separation process directly affects the utilization of tubes, the processing quality of parts, production efficiency and production costs. For example, the material preparation process of cold extrusion parts, inner and outer rings of bearings and sleeves of metal chains is inseparable from the cutting and separation process of tubes, and the amount of parts is amazing. Due to the hollow structure, the tube is easily deformed due to radial impact and compression. Currently, common tube separation methods include lathes and saw blades. When turning, a large amount of raw materials need to be turned off. When cutting with saw blades, the saw blades will be lost too quickly, the cost will be too high and it will also cause significant waste of raw materials and high noise [1–7]. Despite the low cost of shear separation, the produced tube blanks have low precision, which cannot be directly used in precision forming processes [8–10]. At present, there are many kinds of precision separation methods. However, there is a lack of research on tube precision separation, only studies such as shearing with a core bar [11,12]. In addition, this separation method will cause inevitable defects such as collapse and tear of tube section. Some separation methods

with high energy consumption, such as laser cutting, can only cut tubes of medium and small thicknesses, and the efficiency and cost of cutting thick-walled tubes are low. There is also the problem that the hardness of the cutting surface increases and affects subsequent processing [4,5]. Therefore, the development concept of today's green manufacturing and efficient and precise forming urgently needs a new method for efficient and precise separation of tubes to provide technical support. Therefore, it is necessary to further explore and improve the new tube precision separation technology and separation mechanism.

The pneumatic fatigue impact loading precision separation process proposed in this paper is a near-net shape separation method. Based on the prefabricated annular notch stress concentration effect and radial repeated impact load on the surface of the tube, this method can control the crack initiation and propagation of the annular notch, thereby realizing tube precision separation. With a GCr15 thick-walled steel tube as the research object, this paper combines the fatigue fracture principle with experimental research to study the influence of fatigue load control on the initiation and expansion of fatigue crack and tube fracture quality during tube separation.

## 2. Principle of Pneumatic Impact Loading Separation

The thick-walled tube's pneumatic fatigue impact loading precision separation process comprehensively utilizes the notch effect and stress concentration effect of the prefabricated annular V-notch, and the expansion mechanism of fatigue crack. For tube precision separation, the main basis for evaluating the separation method is the fracture quality and separation efficiency of the lower blank. In order to obtain ideal fracture quality and separation efficiency, the key technology of this separation method lies in the effective utilization of fatigue crack and the controllable expansion of fatigue crack. By rapidly initiating microcracks and promoting their stable growth, high-quality precision separation is finally achieved. The working principle of the pneumatic fatigue impact loading precision separation method is shown in Figure 1.

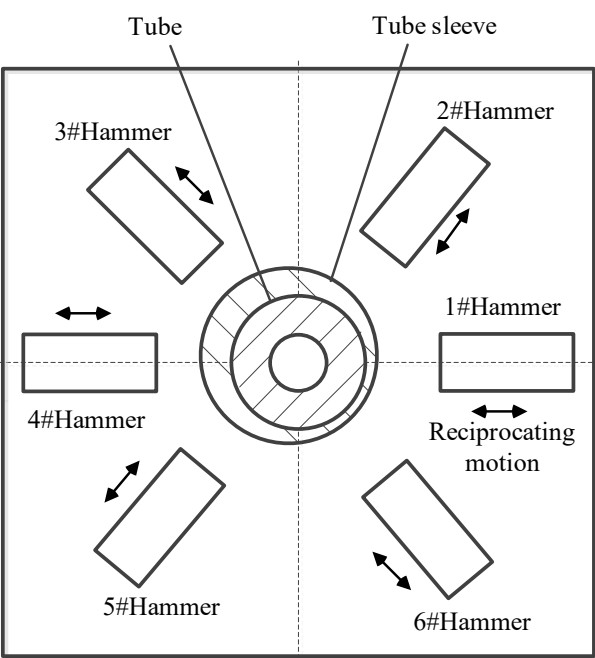

**Figure 1.** Working principle of pneumatic fatigue impact separation.

First of all, the lathe and other processing machines turn the annular V-notch with specific geometric parameters on the specific position of the tube raw material. The annular V-notch produces the desired notch effect in the local area of root. Then, one end of the prefabricated annular V-notch metal tube is fixed on the separator tube fixing device, and

the other end is fed into the tube sleeve for a certain distance, in a cantilever state. On the outer circumference of the other end of the tube sleeve, there are six impact hammers evenly distributed. Each impact hammer is driven by an independent cylinder and piston. By changing the pressure of the fluid in the cylinder, the magnitude of the flow output load and the speed, the tube sleeve is subjected to a circumferentially variable displacement load.

Under the rapid impact loading, fatigue microcracks are initiated at the root of the V-notch. Once microcracks are initiated, a source of stress concentration is generated, which will accelerate the initiation and propagation of fatigue cracks. Finally, the separation device performs continuous strikes with a specific load strike order and load amplitude, so that the microstructure crack group grows stably and rapidly and forms a macro fatigue crack. The fatigue crack continues to expand along a certain expansion direction and toward the center point of the tube section until the residual metal cannot withstand the continuous action of the external load and eventually breaks. One precision separation is done.

## 3. Thick-Walled Tube V-Notch Bottom Crack Initiation

The crack initiation of the thick-walled tube V-notch bottom crack is related to the stress intensity factor ($K_I$) at the crack tip and the crack propagation threshold ($\Delta K_{th}$). When $\Delta K_I < \Delta K_{th}$, the crack does not crack. When $\Delta K_I > \Delta K_{th}$, crack initiation and propagation will occur [13–17]. The formula expressed is as follows:

$$\Delta\sigma\sqrt{a_0}Y > \Delta K_{th} \tag{1}$$

where $\Delta\sigma$ is the fatigue stress amplitude, $a_0$ is the crack length, $Y$ is the geometry-dependent correction factor, and $\Delta K_{th}$ is the fatigue crack propagation threshold.

The schematic diagram of tube bending moment is shown in Figure 2. In the process of tube separation, the groove depth, opening angle and groove bottom radius of the geometric parameters of the tube V-notch are expressed as $h$, $\alpha$ and $\rho$, respectively; $r$ and $\theta$ are polar coordinates; $\sigma$ is the nominal stress at the tip of the V-notch. Under the plane strain state, when the tube is subjected to the bending load F and the bending moment M, the stress field near the tip of the V-notch is expressed as follows [13]:

$$\begin{cases} \sigma_x = \frac{K_I}{\sqrt{2\pi r}}\left[\cos\frac{\theta}{2}\left(1 - \sin\frac{\theta}{2}\sin\frac{3\theta}{2}\right) - \frac{\rho}{2r}\cos\frac{3\theta}{2}\right] \\ \sigma_y = \frac{K_I}{\sqrt{2\pi r}}\left[\cos\frac{\theta}{2}\left(1 + \sin\frac{\theta}{2}\sin\frac{3\theta}{2}\right) + \frac{\rho}{2r}\cos\frac{3\theta}{2}\right] \\ \sigma_{xy} = \frac{K_I}{\sqrt{2\pi r}}\left[\cos\frac{\theta}{2}\sin\frac{\theta}{2}\cos\frac{3\theta}{2} - \frac{\rho}{2r}\sin\frac{3\theta}{2}\right] \\ \sigma_z = 2\nu\frac{K_I}{\sqrt{2\pi r}}\cos\frac{\theta}{2} \\ \sigma_{xz} = \sigma_{yz} = 0 \end{cases} \tag{2}$$

where $\nu$ is Poisson's ratio. The $K_I$ in Equation (2) is modified to accurately express the stress intensity factor at the tip of the annular V-groove on the surface of the tube. The corrected $K_I$ can be expressed as:

$$K_I = \sigma\sqrt{\pi h}f_I(2h/D) \tag{3}$$

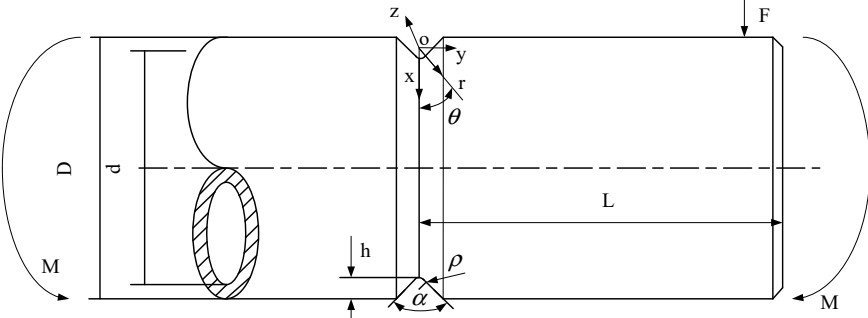

**Figure 2.** Schematic diagram of tube bending moment.

When the stress ratio is $R = 0$,

$$K_I = K_{max} = \Delta K \tag{4}$$

By combining with the above formula,

$$\Delta K = \Delta \sigma \sqrt{\pi h} f_I(2h/D) \tag{5}$$

where $f_I(2h/D)$ is the correction factor related to the geometry and size of the tube and $\Delta K$ is the magnitude of the stress intensity factor. This paper analyzes only the stress magnitude where the $r$ tends to the $\rho/2$, that is, the stress magnitude at the tip of the V-notch. According to fracture mechanics, $\sigma_y$ has the greatest influence on crack propagation in the Type I. Therefore, $r = \rho/2$ and $K_I$ are substituted into Equation (2) to obtain $\sigma_y = \frac{\sigma \sqrt{t_{yx}}}{\sqrt{\rho}} f_I(2h/D)[\cos\frac{\theta}{2}(1 + \sin\frac{\theta}{2}\sin\frac{3\theta}{2}) + \cos\frac{3\theta}{2}]$. When $\theta = 0$, $\sigma_y = 2\sqrt{\frac{t_{yx}}{\rho}} f_I(2h/D)\sigma$. Let $k_\alpha = 2\sqrt{\frac{t_{yx}}{\rho}} f_\alpha(2h/D)$ be the stress concentration factor of the V-notch notch. When $\alpha = 0$,

$$k_0 = 2\sqrt{\frac{t_{yx}}{\rho}} f_0(2h/D) \tag{6}$$

$t_{yx}$ refers to the effective crack length.

When a rod with circumferential cracks is only subjected to bending loads [18]:

$$
\begin{aligned}
f_0(2h/D) = \quad & \frac{3}{8}\sqrt{(1 - \frac{2h}{D})}[1 + \frac{1}{2}(1 - \frac{2h}{D}) + \frac{3}{8}(1 - \frac{2h}{D})^2 \\
& + \frac{5}{16}(1 - \frac{2h}{D})^3 + \frac{35}{128}(1 - \frac{2h}{D})^4 + 0.537(1 - \frac{2h}{D})^5]
\end{aligned}
\tag{7}
$$

When the V-notch opening angle is $\alpha$, its theoretical stress concentration factor can be obtained [18]:

$$k_\alpha = 1 + (k_0 - 1) \times [1 - (\frac{\alpha}{180})^{1 + 2.4\sqrt{\rho/t_{yx}}}] \tag{8}$$

When $k_\alpha$ and $k_0$ are substituted into Equation (5), it can get

$$f_\alpha(2h/D) = \frac{1}{2}\sqrt{\frac{\rho}{t_{yx}}} + [f_0(2h/D) - \frac{1}{2}\sqrt{\frac{\rho}{t_{yx}}}] \times [1 - (\frac{\alpha}{180})^{1 + 2.4\sqrt{\rho/h}}] \tag{9}$$

In Equation (5), $\Delta \sigma$ is the nominal stress of the tube at the tip of the V-notch and its expression is

$$\Delta \sigma = \frac{4\Delta M}{\pi a_0^3} \tag{10}$$

Force on the tube:

$$F = md(2\pi f)^2 \sin w t_s \tag{11}$$

Tube bending moment:

$$\Delta M = \Delta F L = 2m(2\pi f)^2 dL \tag{12}$$

where $m$ is the mass of the blanking hammer and blanking die; $f$ is the frequency of applying impact load; $t_s$ is the time; $\Delta F$ is the impact force on the tube; and $L$ is the length of the arm when the tube is impacted. Equations (9) and (12) are substituted into Equation (5) and compared with Equation (1). Initiation and expansion of root fatigue crack of tube V-notch were analyzed and judged. It can be seen from Equation (12) that the bending moment of the tube during the impact loading process is related to the mass of the impact hammer and the frequency of the impact load. However, only the frequency of the impact load is a variable, so in the process of separation experiment research, mainly starting from adjusting the frequency of impact load, many tests and analyses were carried

out. Therefore, the regulation of the impact load size and loading frequency of the tube can control the initiation and expansion of cracks. The relationship between impact load and crack initiation time in this paper is shown in Figure 3.

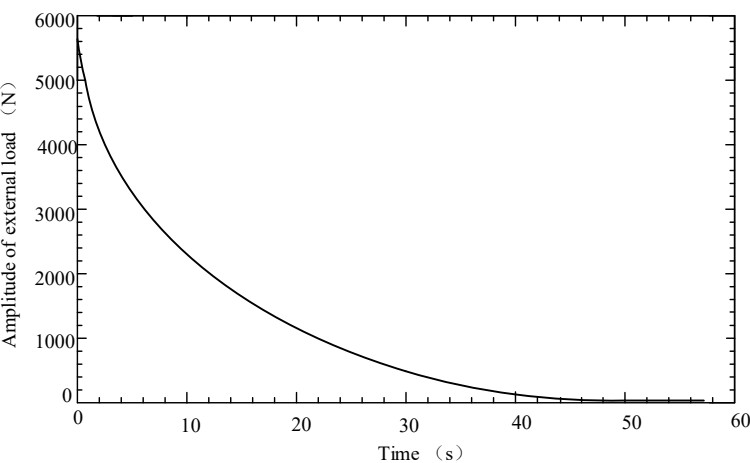

**Figure 3.** Relationship between external load amplitude and crack initiation time.

## 4. Research on the Separation Experiment of Pneumatic Fatigue Impact Loading

### 4.1. Experimental Setup

The pneumatic fatigue impact loading precision separation device is shown in Figure 4. The experimental device is mainly composed of six double-acting, single-piston striking cylinders evenly distributed in the circumference of the cylinder body and six striking hammer heads connected to the cylinder piston. The cylinder diameter of the six single-piston cylinders is 180 mm, the diameter of the piston rod in the cylinder is 130 mm and the stroke of the cylinder is 10 mm, which can realize the separation of tubes of different materials, diameters and wall thicknesses. The use of the sleeve structure can realize the separation of the small aspect ratio tube.

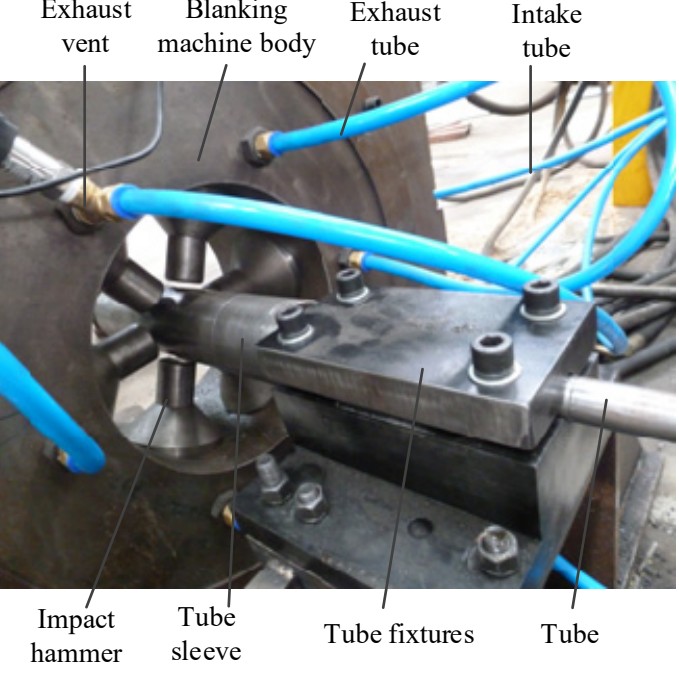

**Figure 4.** Fatigue impact loading separation test setup.

The centerline of the clamping device and the separation device are at the same level. The load arm of the tube sleeve is lengthened, which increases the bending moment. The tube sleeve is placed between the separation device and the clamping device. The sleeve bracket of elastic material is installed inside the tube for fixed installation. Among them, the damping effect of the elastic material enables the tube sleeve to automatically reset after each load. The reduced diameter piston rod forms an independent retraction cavity and feed cavity in each cylinder. Combined with independent intake and exhaust circuits, the feeding and retracting of the hammer head is completed. Therefore, the adjustment of the working air pressure is equal to the adjustment of the impact load.

The geometrical parameters of the experimental tube are shown in Figure 5, including the tube separation length L1, the outer diameter D of the tube, the inner diameter d and the geometric parameters of the notch. In the tube separation experiment, the stress concentration groove on the tube surface adopts the annular V-notch. The geometric parameters of annular V-notch include notch flare angle $\alpha$, notch depth h and notch root base angle radius $\rho$.

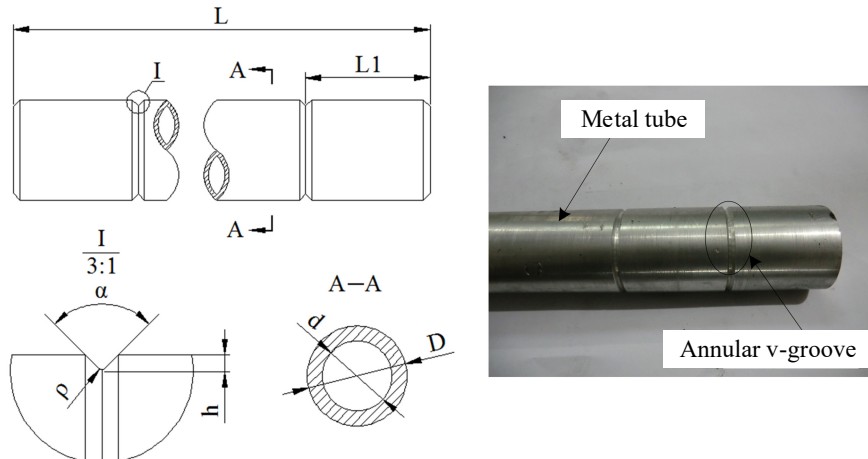

**Figure 5.** Schematic diagram and physical photos of specimen geometric parameters.

The separation test uses $\phi 30 \times 6$ GCr15 steel seamless steel tube, where the outer diameter = 30 mm, the wall thickness = 6 mm and $L1$ = 50 mm. The geometric parameters of the V-notch are taken as $\rho$ = 0.2 mm, $\alpha$ = 90° and $h$ = 1.2 mm, as shown in Table 1.

**Table 1.** Test tube materials and parameters.

| Material | Model | D (mm) | d (mm) | L (mm) | L1 (mm) | ρ (mm) | α (°) | h (mm) |
|---|---|---|---|---|---|---|---|---|
| GCr15 | $\phi 30 \times 6$ | 30 | 18 | 1000 | 50 | 0.2 | 90° | 1.2 |

### 4.2. Control Curve of Tube Precision Separation

The tube V-notch root crack will expand with a linear decrease in the hammer strike frequency and a linear increase in the radial displacement. When the average stress ratio applied to the tube increased, the loading frequency of impact load showed an overall trend from high to low. In the shortest time, efficient and high-quality tube separation is achieved [19–22]. Based on the separator control system, the frequency time control curve of the separator was designed in this experiment to control the loading frequency of the impact load, as shown in Figure 6. In the multistep decline frequency-time control curve, $f$ is the constant loading frequency, $t_c$ is the total time taken to complete the separation, $f_0$ is the initial frequency, $f_1$ is the terminate frequency, $f_c$ is the corresponding frequency when the separation is completed, $\Delta f$ is the amount of change in the stage frequency and $\Delta t$ is the frequency duration.

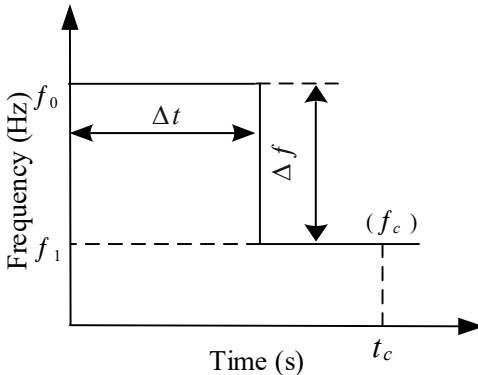

**Figure 6.** Multistep decline frequency time control curve.

For the fatigue impact loading precision separation test, a multistep decline frequency time control curve is used. The specific experimental parameters of the control curve are shown in Table 2. The test numbers I, II, III and IV in Table 2 represent four tests conducted independently, respectively. It should be noted that different test parameters were used in each test.

**Table 2.** GCr15 steel tube separation experimental parameters.

| Test Number | $f_0$ (Hz) | $f_1$ (Hz) | $\Delta f$ (Hz) | $\Delta t$ (s) |
|:---:|:---:|:---:|:---:|:---:|
| I | 16.5 | 4.5 | 2 | 4 |
| II | 20 | 4 | 2 | 4 |
| III | 25.5 | 3.5 | 2 | 4 |
| IV | 31 | 3 | 2 | 4 |

In the four tests, the initial frequency $f_0$ is 15 Hz, 20 Hz, 25 Hz and 30 Hz, and the termination frequency $f_1$ is 4.5 Hz, 4 Hz, 3.5 Hz and 3 Hz, respectively. In addition, the phase frequency change amount $\Delta f$ is all 2 Hz and the frequency duration $\Delta t$ is all 4 s.

### 4.3. Test Results and Discussion

The separation test used the multistep decline frequency time control curve shown in Figure 6. The blank photos obtained from the four groups of separation tests are shown in Figure 7.

In order to further analyze the cross-section characteristics of the thick-walled tube under the aerodynamic fatigue impact loading of the precision separation process, a schematic diagram of the fatigue fracture characteristics of the tube was drawn, and the fatigue fracture of the thick-walled tube was observed and analyzed using a scanning electron microscope (SEM), as shown in Figure 8. Figure 8a identifies the three most important growth regions during the separation of thick-walled tubes: the crack initiation area, the fatigue crack growth region, and the transient break region. Figure 8b shows the fracture morphology of the crack initiation area magnified by 500 times. It can be seen from the figure that the crack initiation area is located on the outer surface of the tube, and there are defects such as impurities and pores in the crack initiation area; Figure 8c is the surface morphology of the fatigue crack extension zone when magnified by 1000 times, and the fracture surface has obvious fatigue arc characteristics; Figure 8d is the micro-morphology of the instantaneous fracture zone when magnified by 500 times. It can be seen that the fracture in the instantaneous fracture zone is a dimple shape, which is caused by the tearing of the tube when it is momentarily broken.

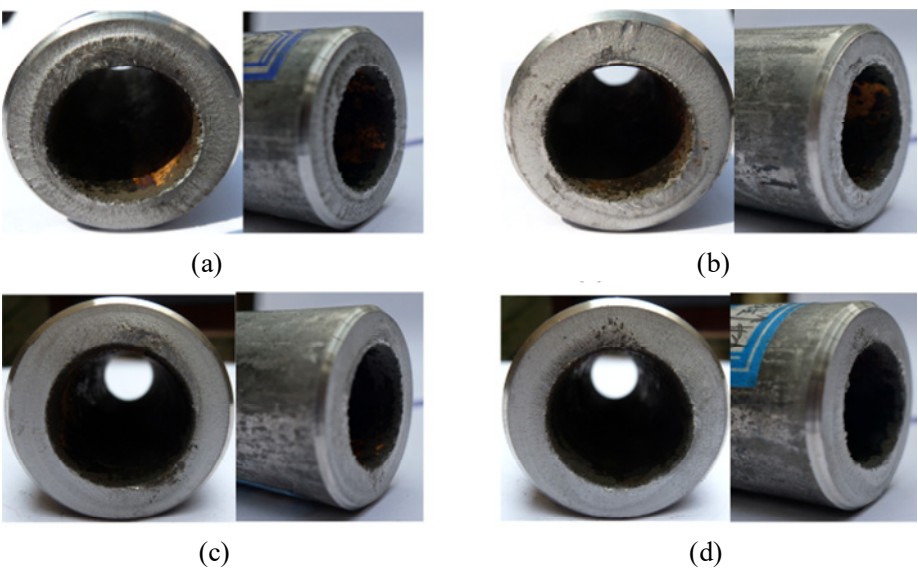

**Figure 7.** Tube section obtained from tests. (**a**) Section of tube obtained in test I; (**b**) Section of tube obtained in test II; (**c**) Section of tube obtained in test III; (**d**) Section of tube obtained in test IV.

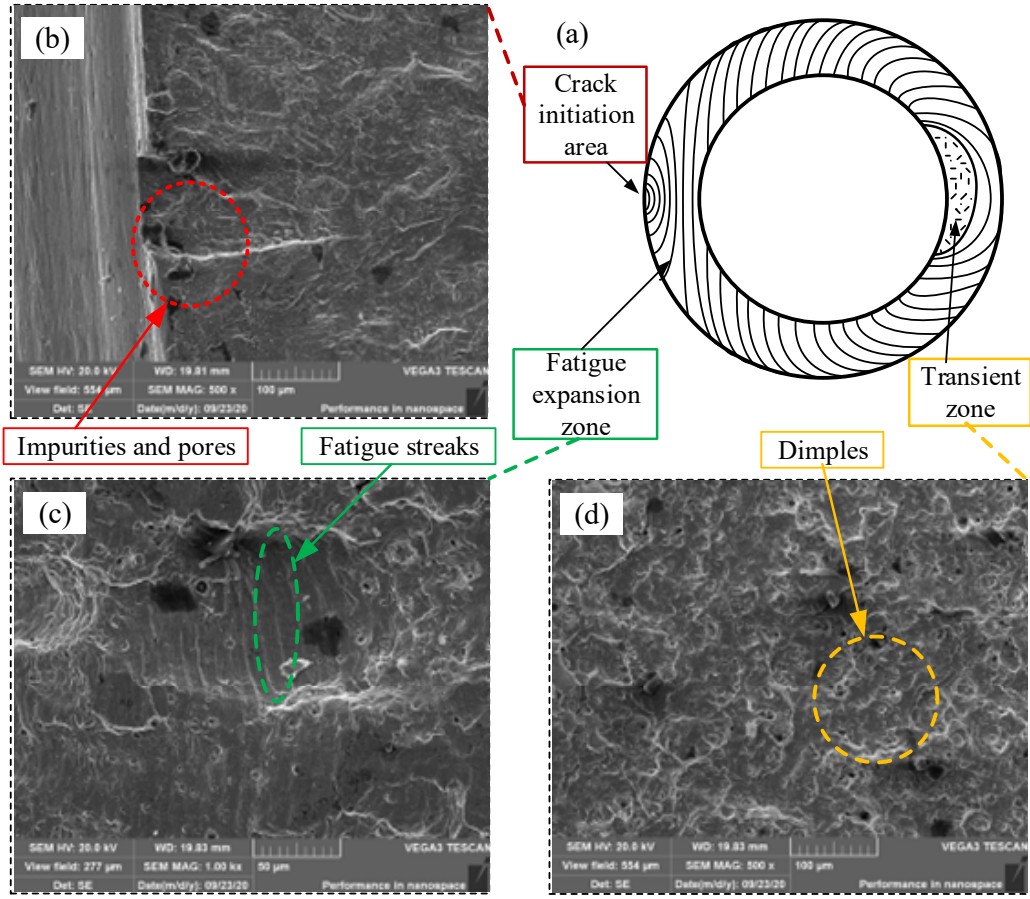

**Figure 8.** Microstructure of fatigue fracture of thick-walled tube: (**a**) Schematic diagram of the division of fatigue fracture zones; (**b**) 500 times magnification of the crack initiation area; (**c**) 1000 times magnification of the crack propagation area; (**d**) 500 times magnification of the transient region.

In order to reasonably evaluate the fracture quality of the tube blank obtained by separation, a fatigue impact precision separation blank fracture quality evaluation method is proposed. The specific parameters of the evaluation method are shown in Figure 9.

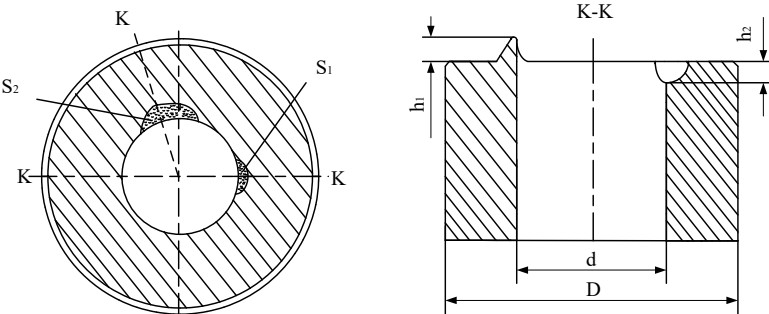

**Figure 9.** Schematic diagram of the evaluation method for tube fracture quality.

In Figure 9, D is the tube outer diameter and d is the tube inner diameter. The maximum height value of the tube section's bulge and the maximum depth value of the pit are represented by $h_1$ and $h_2$, respectively, which can be measured with conventional measuring tools, such as length measuring instruments. The convex area and the concave area of the tube section are denoted by $S_1$ and $S_2$, respectively. $a_1\%$ and $a_2\%$ are the percentages of the convex area ($S_1$) and the concave area ($S_2$) of the entire tube area, which can be calculated based on the measurement data of the section planeness. With the aid of a 3D coordinate measuring machine, the planeness of the section is measured, which can be used for an intuitive evaluation of the characteristics of the section.

In addition to the evaluation of blank fracture quality, the comprehensive scientific evaluation of the overall performance of the separation needs to increase the evaluation index separation completion time $t_c$. Table 3 shows the section parameters and separation time of the GCr15 steel tube. Furthermore, the experimental numbers I, II, III and IV in Table 3 correspond to the tube cross-sectional picture numbers (a), (b), (c) and (d) in Figure 7, respectively.

**Table 3.** Section parameters and separation time of the GCr15 steel tube.

| Test Number | $h_1$ (mm) | $h_2$ (mm) | $a_1$ (%) | $a_2$ (%) | $t_c$ (s) |
|---|---|---|---|---|---|
| I | 0.76 | 0.16 | 13.6 | 0.15 | 45 |
| II | 0.68 | 0.13 | 12.5 | 0.12 | 33 |
| III | 0.22 | 0.15 | 0.21 | 0.09 | 35 |
| IV | 0.17 | 0.12 | 0.20 | 0.08 | 20 |

According to the experimental blank photo in Figure 7 and the experimental result data in Table 3, the initiation and expansion of cracks were effectively controlled in the separation experiment of the GCr15 seamless steel tube (30 mm in diameter and 6 mm in wall thickness). Among them, experiment IV had the best effect. The maximum height of the protrusion of the tube section is 0.17 mm, and the maximum depth of the pit is 0.12 mm. The area of the convex area and the area of the concave area account for only 0.20% and 0.08% of the entire section, respectively. According to the above data, it can be found that a relatively flat tube blank section is obtained. In order to visually judge the tube fracture quality on a macroscopic level, a 3D coordinate measuring machine was used to measure the planeness of the tube section shown in Figure 7a,d. The measured data were processed by MATLAB software (version 9.4, MathWorks, Natick, MA, USA). Then, a 3D map of the planeness values of the cross-section of the GCr15 seamless steel tube was drawn, as shown in Figure 10a,b. The specific value can reflect the planeness of the tube section. The color of the color bar of 3D plot represents the numerical value of the convexity and concavity of the tube section, in millimeters. In this 3D map, the photo of the blank obtained in the experiment (Figure 7a) can be seen more intuitively. The quality of the section in the instantaneous fracture area is relatively poor, which is obviously caused by the tearing phenomenon caused by the high termination frequency, and the quality of the section in the crack propagation area is not much different, being relatively flat and of better quality.

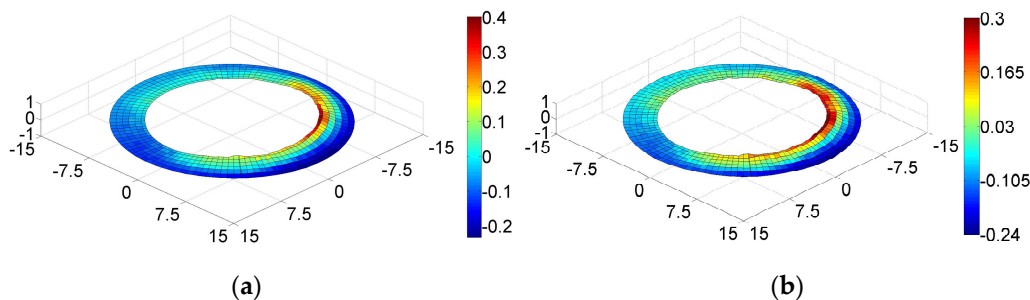

**Figure 10.** Three-dimensional view of the GCr15 seamless steel tube section. (**a**) Cross-sectional flatness value for test I; (**b**) Cross-sectional flatness value for test IV.

Due to the extended frequency dwell time, the crack obtained a stable propagation time at each stage. Correspondingly, the frequency also has sufficient dwell time in the crack propagation stage. Due to the stable mutual extrusion between the crack surfaces, there is no unevenness near the blank section. The crack propagation area of the entire blank section is smooth and clean, and there is almost no final fracture region. Overall, the fracture quality was good. Thus, it can be concluded that the frequency continuous multistep decline control curve has higher efficiency and better blank fracture quality.

In experiment I, because the value of the control curve's terminate frequency $f_1$ is too large, the final fracture region of the section is more obvious, which produces a narrow and long instantaneous fracture zone. Experiments III and IV are adjusted accordingly. By appropriately reducing the end frequency, the effect of fracture quality is significantly improved. In particular, the fracture quality of the IV test was very high and the separation time was relatively short. Therefore, it can be concluded that the value of the terminate frequency $f_1$ of the control curve cannot be too large. In addition, the sudden change of the loading frequency for many times during the expansion process will have a certain adverse effect on the expansion of the crack, such as the unevenness at the inner ring of the blank section. Therefore, the follow-up experimental research needs to be optimized accordingly.

The fatigue impact loading precision separation method was used to obtain high quality tube blank sections. This requires that the ratio of the area of the tube final fracture region of the entire cross-sectional area is as small as possible, and the maximum concave-convex size is as small as possible. According to the previous analysis, the realization of tube precision separation needs to study the parameters in the control curve. The effects of initial frequency $f_0$ and terminate frequency $f_1$ on fracture quality are analyzed. In the case of the same initial frequency $f_0$, the influence of multiple groups of different termination frequencies $f_1$ on tube blank fracture quality is analyzed. In the case of the same termination frequency $f_1$, the influence of multiple groups of different initial frequency $f_0$ on tube blank fracture quality is analyzed.

The initial frequency $f_0$ used ranges from 22 Hz to 4 Hz, and the frequency duration of each stage is 4 s. The terminate frequency $f_1$ is set to 2 Hz. The relationship between initial frequency and tube fracture quality is shown in Figure 11a. The initial frequency $f_0$ corresponding to all the termination frequencies $f_1$ here is 13 Hz as the value of the experimental study, and the frequency duration of each stage is 4 s. The value range of terminate frequency is 1 Hz~8.5 Hz. The relationship between the termination frequency and tube fracture quality is shown in Figure 11b.

It can be seen from Figure 11a that the area of the final fracture region decreases by less than 2% in the process of increasing the initial frequency from 4 Hz to 22 Hz. This shows that the initial frequency $f_0$ has a certain influence on the percentage of tube blank final fracture region, but it is not significant. It can be seen from Figure 11b that the larger the terminate frequency $f_1$, the larger the tube blank final fracture region, especially when the $f_1$ is greater than 3.5 Hz. In the process of increasing the termination loading frequency a from 1 Hz to 8.5 Hz, the area of the transient zone increased by 21%. As the influence on

tube blank fracture quality becomes more and more obvious, the percentage of tube blank final fracture region gradually increases.

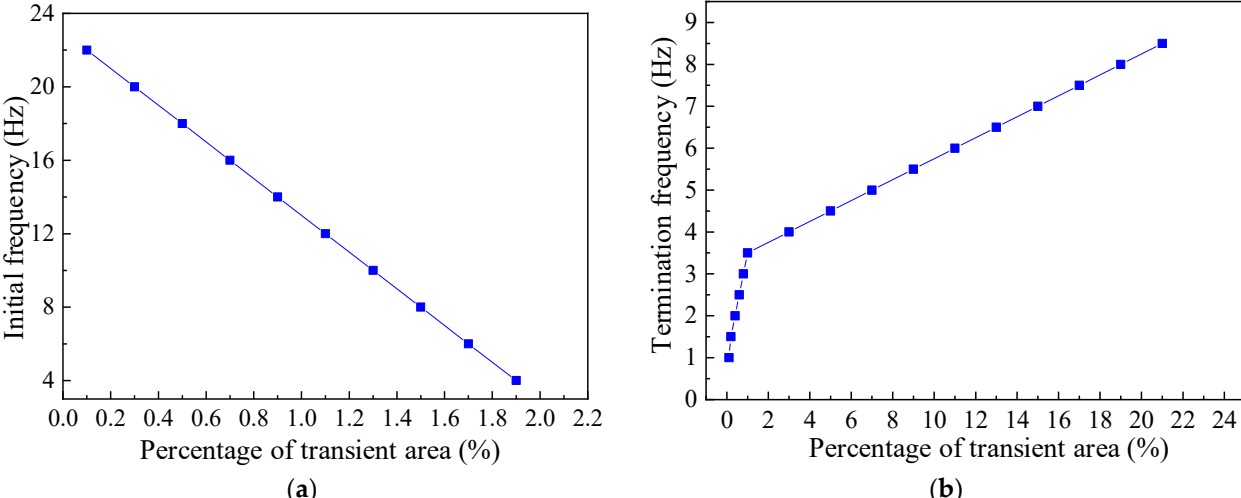

(**a**) (**b**)

**Figure 11.** The relationship between initial frequency, termination frequency and fracture quality: (**a**) Relationship between $f_0$ and fracture quality; (**b**) Relationship between $f_1$ and fracture quality.

The tube separation sections obtained by different tube separation methods are obviously different. Figure 12 is a comparison of the tube cross-section obtained by the new process (tube fatigue impact loading precision separation) and the traditional process (rotary wedge method and impact shear method). It can be seen from Table 4 that the tube blank sections obtained by the novel precision separation method proposed in this paper have higher geometric accuracy, verticality and planeness. With negligible geometric deformations and defects such as burrs, the tube can be directly used in the production and processing of parts such as bearing rings, chain rollers and sleeves. The tube blank section obtained by the traditional shear separation method generally has defects such as horseshoe deformation and burrs. Such blanks need to go through the process of dealing with defects before they can be used in the production of parts. At the same time, the new tube separation device significantly reduces material loss and improves production efficiency.

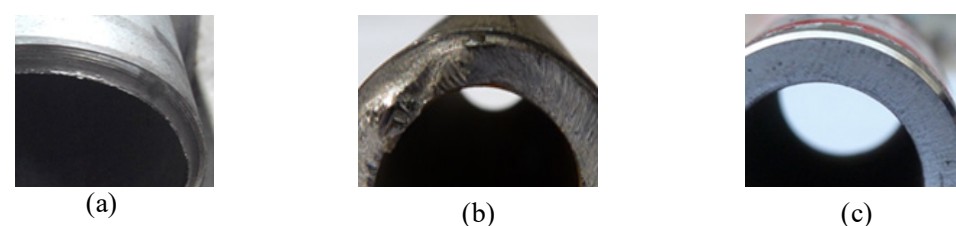

(a) (b) (c)

**Figure 12.** Comparison of tube sections obtained by different separation methods: (**a**) Rotary wedge shear method; (**b**) Impact shear method; (**c**) Fatigue impact loading separation method for a thick-walled tube.

**Table 4.** Comparison of parameters of tube sections obtained by different separation methods.

| Separation Method | Fractional Defect Ratio (%) | Section Verticality | Section Flatness (mm) |
| --- | --- | --- | --- |
| Rotary wedge shear method | 7.9 | 0.3 | 1.0 |
| Impact shear method | 12.5 | 0.5 | 3.5 |
| Fatigue impact loading separation method for a thick-walled tube | 1.9 | 0.01 | 0.8 |

### 5. Conclusions

(1)  The thick-walled tube pneumatic fatigue impact loading precision separation process proposed in this paper effectively combines radial cyclic impact load, the stress concentration effect and the fatigue fracture principle, which realizes the thick-walled tube fatigue fracture.

(2)  In the analysis of crack initiation at the bottom of the V-notch in thick-walled tube, the threshold value of crack initiation at the tip of the V-notch was analyzed when the tube was subjected to bending load F and bending moment M under a plane strain state, and the expression of the corresponding theoretical stress concentration coefficient was obtained when the V-notch opening angle was $\alpha$ and there was a relationship between the impact load and the loading frequency of the tube and the bending moment. Based on this, it can be judged whether the fatigue crack at the root of the tube V-notch was cracked.

(3)  Under the cyclic action of pneumatic impact load, the control method of gradually reducing the loading frequency from high to low can effectively control the initiation, stable expansion and rapid rupture of thick-walled tube cracks. Implements' tube fatigue fracture and combining with thick wall tube fatigue fracture microscopic morphology, fracture analysis of the thick wall tube in the section of pneumatic impact fatigue loading precision separation characteristics of the GCr15 seamless steel tube is obtained by measuring and data processing sections. Figure 3D data shows a contrast analysis of the initial frequency and termination frequency of the relationship with the fracture quality. According to the proposed fracture quality evaluation method, the fracture quality of the separation section of the thick-walled tube is effectively analyzed.

(4)  The separation device of pneumatic fatigue impact loading precision separation process has a simple structure, is easy to operate, saves raw materials, has low energy consumption and obtains a good thick-walled tube precision separation effect, which has certain engineering application significance.

**Author Contributions:** Conceptualization and methodology, R.-F.Z., W.-C.G., D.-Y.Z. and X.-D.X.; Formal analysis, R.-F.Z. and Y.-W.L.; Supervision, R.-F.Z. and W.-C.G.; Writing—original draft preparation, R.-F.Z.; Writing—review and editing, W.-C.G. and R.-Z.P. All authors have read and agreed to the published version of the manuscript.

**Funding:** This research has been supported by the China Postdoctoral Science Foundation (No. 2018M633543) and Shaanxi Key Laboratory of Mine Electromechanical Equipment Intelligent Monitoring (No. SKL-MEEIM201908).

**Institutional Review Board Statement:** Not applicable.

**Informed Consent Statement:** Not applicable.

**Data Availability Statement:** The data presented in this study are available on request from the corresponding author.

**Conflicts of Interest:** The authors declare no conflict of interest.

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
