# Peer review of "New Technology and Experimental Research on Thick-Walled Tube Fatigue Impact Loading Precision Separation"

_metals, doi:10.3390/met12050837_

Round 1

Reviewer 1 Report

  1. In the abstract, it would be better to briefly summarize about the experimental setup and control curve and add them.
  2. 1) It is necessary to accurately classify and fill in the information in Table 1.

2) Indicate the pipe dimensions and units accurately.

3) Enter to be separated by dividing row and column the geometric parameters of V-notch.

4) Enter the value of h in Table 1.

  1. 1) There is no information about test number Ⅳ in table

2) In Table 2, ‘Test number’ and ‘f0’ are duplicated.

  1. It is not confusing to indicate units in the Table 1, 2 and 3 with parentheses instead of slashes.
  2. 1) Does the crack source (where the crack growth begins) indicate crack initiation?

The contents in parentheses indicate as "where crack growth begins".

Strictly speaking, it is desirable to distinguish between crack initiation and crack growth.

2) Indicate what the markings in Fig. 7 indicate, respectively.

  1. There is a need for unity of terminology.

- initial loading frequency or initial frequency

- terminate loading frequency or termination frequency

  1. “The initial loading frequency is set to 13Hz, and the frequency duration of each stage is 4s.”

Now what is this sentence trying to say? I don't understand.

  1. From “When the termination loading frequency = 5.5Hz, the area of tube blank final fracture region is close to 10%.”, what does 10% mean?

That is, why did you mention 10% of the area of the final fracture region?

  1. From “It can be seen from Fig. 10(b) that the area of the final fracture region decreases by less than 2% in the process of increasing the initial loading frequency from 4 Hz to 22 Hz.”

The initial loading frequency is not shown in the Fig. 10(b).

  1. Although it should clearly indicate how the formulas presented in this study were used in the results of the study, there is no content.

In particular, it is unclear how fatigue cracking, which was presented as a core technology of the separation method, was utilized.

How the parameters presented in the formula were used to obtain the research results should be expressed as quantitative values through graphs, etc.

  1. In Fig. 11, the differences from other separation methods are simply compared and explained using the fracture surface.

At this time, geometric accuracy, verticality, and planeness are mentioned, but there is no comparative data for them. Supplementation is needed.

  1. 1) It is necessary to rewrite the conclusion.

In particular, in (1) and (2), the contents of the research results related to the main text are insufficient.

2) The results of the stress field analysis are not in the text.

And, the content of "the stress concentration effect of the V-notch on the tube surface and the fatigue crack initiation and expansion mechanism" is not clear or has no related content in the text.

  1. I think it is good to try a new separation method in this study.

However, as mentioned above, it is judged that there are many insufficiencies in the results of the study in general, and the discussion of the results is also insufficient.

I think it is necessary to further improve the completeness of the paper.

Author Response

1.In the abstract, it would be better to briefly summarize about the experimental setup and control curve and add them.

Authors’ response:

Thanks a lot for the reviewer’s comments.

The experimental setup and control curves have been summarized and added to the abstract of the paper.

2.1) It is necessary to accurately classify and fill in the information in Table 1.

2) Indicate the pipe dimensions and units accurately.

3) Enter to be separated by dividing row and column the geometric parameters of V-notch.

4) Enter the value of h in Table 1.

Authors’ response:

Thanks a lot for the reviewer’s comments.

We apologize for this ignorance, (1), (2), (3), (4) have been corrected and we have checked and corrected all the underlying mistakes in the full text.

Table 1. Test tube materials and parameters

Material

Model

D(mm)

d(mm)

L(mm)

L1(mm)

(mm)

(o)

h(mm)

GCr15

30×6

30

18

1000

50

0.2

90o

1.2

3、1) There is no information about test number Ⅳ in table

2) In Table 2, ‘Test number’ and ‘f0’ are duplicated.

Authors’ response:

Thanks a lot for the reviewer’s comments.

We apologize for this ignorance, (1), (2) have been corrected and we have checked and corrected all the underlying mistakes in the full text.

Table 2. GCr15 steel tube separation experimental parameters

Test number

(Hz)

(Hz)

(Hz)

(s)

16.5

4.5

2

4

20

4

2

4

25.5

3.5

2

4

31

3

2

4

4、It is not confusing to indicate units in the Table 1, 2 and 3 with parentheses instead of slashes.

Authors’ response:

Thanks a lot for the reviewer’s comments.

The units in Tables 1, 2 and 3 have been represented in the text by parentheses instead of slashes.

5、1) Does the crack source (where the crack growth begins) indicate crack initiation?

The contents in parentheses indicate as "where crack growth begins".

Strictly speaking, it is desirable to distinguish between crack initiation and crack growth.

Authors’ response:

Thanks a lot for the reviewer’s comments.

The source of the crack (where crack growth begins) is the indication of crack initiation.

The contents in parentheses indicate as "where crack growth begins" has been removed from the text.

2) Indicate what the markings in Fig. 7 indicate, respectively.

Authors’ response:

Thanks a lot for the reviewer’s comments.

What the mark represents has been written in Fig. 7.

6、There is a need for unity of terminology.

- initial loading frequency or initial frequency

- terminate loading frequency or termination frequency

Authors’ response:

Thanks a lot for the reviewer’s comments.

The initial loading frequency and termination loading frequency in the text have been unified into the initial frequency and termination frequency.

7、“The initial loading frequency is set to 13Hz, and the frequency duration of each stage is 4s.”

Now what is this sentence trying to say? I don't understand.

Authors’ response:

Thanks a lot for the reviewer’s comments.

The initial frequency is the frequency at which the tube is initially loaded during the experiment, and the frequency duration refers to the time for continuous loading while keeping the frequency value unchanged at this frequency. As shown in the figure, is the frequency continuous loading time. And the typo in the text "frequency continuous loading time " has been changed to "frequency continuous loading time ". Figure 10(b) shows the relationship between the stop frequency and the cross section. If there are multiple stop frequencies, there should be corresponding initial frequencies. The initial frequencies corresponding to all stop frequencies here are selected as 13Hz as the value of the experimental study, only It is only the initial frequency corresponding to the stop frequency, and has no special meaning.

8、From “When the termination loading frequency = 5.5Hz, the area of tube blank final fracture region is close to 10%.”, what does 10% mean?

That is, why did you mention 10% of the area of the final fracture region?

Authors’ response:

Thanks a lot for the reviewer’s comments.

The meaning of this sentence in the original text is: with the increase of the termination frequency, the impact on the quality of the tube blank section becomes larger and larger. When the termination frequency =5.5Hz, the instantaneous break area of the pipe blank section is close to 10%, has reached a large proportion. Since the meaning is not clear, the original expression has been revised, and "When the termination loading frequency = 5.5Hz, the area of tube blank final fracture region is close to 10%." is changed to " In the process of increasing the termination loading frequency a from 1 Hz to 8.5 Hz, the area of the transient zone increased by 21%.”.

9、From “It can be seen from Fig. 10(b) that the area of the final fracture region decreases by less than 2% in the process of increasing the initial loading frequency from 4 Hz to 22 Hz.”

The initial loading frequency is not shown in the Fig. 10(b).

Authors’ response:

Thanks a lot for the reviewer’s comments.

The statement is wrong, "It should be seen from Fig. 10(a) that the area of the transient area decreases by less than 2% when the initial frequency increases from 4Hz to 22Hz," and Fig. 10(b) Modified to Fig. 10(a).

10、Although it should clearly indicate how the formulas presented in this study were used in the results of the study, there is no content.

In particular, it is unclear how fatigue cracking, which was presented as a core technology of the separation method, was utilized.

How the parameters presented in the formula were used to obtain the research results should be expressed as quantitative values through graphs, etc.

Authors’ response:

Thanks a lot for the reviewer’s comments.

(1) It can be seen from Equation 12 that the bending moment of the tube during the impact loading process is related to the mass of the impact hammer and the frequency of the impact load. However, only the frequency of the impact load is a variable, so this paper focuses on the pipe material. In the process of separation experiment research, mainly starting from adjusting the frequency of impact load, many tests and analyses were carried out.

(2) If it is necessary to analyze whether the fatigue crack starts or not, it can be analyzed with the formula derived theoretically in this paper. The specific judgment method is to substitute formulas (9) and (12) into formula (5) and compare with (1), then determine whether the fatigue crack at the root of the V-notch of the tube is cracked.

(3) The relationship between the magnitude of the impact load and the crack initiation in this paper is shown in the figure:

11、In Fig. 11, the differences from other separation methods are simply compared and explained using the fracture surface.

At this time, geometric accuracy, verticality, and planeness are mentioned, but there is no comparative data for them. Supplementation is needed.

Authors’ response:

Thanks a lot for the reviewer’s comments.

We have given the data of the geometric accuracy, verticality and flatness of the tube section in the text, and put the data of different separation methods in Table 4 for comparison.

Table 4. Comparison of parameters of tube sections obtained by different separation methods

Separation method

Fractional defect ratio (%)

Section verticality

Section flatness (mm)

Rotary wedge shear method

7.9

0.3

1.0

Impact shear method

12.5

0.5

3.5

Fatigue impact loading separation method for thick-walled tube

1.9

0.01

0.8

12、1) It is necessary to rewrite the conclusion.

In particular, in (1) and (2), the contents of the research results related to the main text are insufficient.

Authors’ response:

Thanks a lot for the reviewer’s comments.

The author has revised the "Conclusion" of this article to read:

(1) The thick-walled tube pneumatic fatigue impact loading precision separation process proposed in this paper effectively combines radial cyclic impact load, stress concentration effect and fatigue fracture principle, which realizes the thick-walled tube fatigue fracture.

(2) In the analysis of crack initiation at the bottom of V-notch in thick-walled tube, the threshold value of crack initiation at the tip of V-notch was analyzed when the tube was subjected to bending load F and bending moment M under plane strain state, and the expression of the corresponding theoretical stress concentration coefficient was obtained when the V-notch opening angle was . And the relationship between the impact load and the loading frequency of the tube and the bending moment. Based on this, it can be judged whether the fatigue crack at the root of the tube V-notch is cracked.

2) The results of the stress field analysis are not in the text.

And, the content of "the stress concentration effect of the V-notch on the tube surface and the fatigue crack initiation and expansion mechanism" is not clear or has no related content in the text.

Authors’ response:

Thanks a lot for the reviewer’s comments.

In the section "2 Initiation of cracks at the bottom of the V-notch of thick-walled tubes", the stress field of the tube under load during the loading process is briefly described. Formula (2) is the stress near the tip of the V-notch. system of field equations.

The "the stress concentration effect of the V-notch on the tube surface and the fatigue crack initiation and expansion mechanism" in the conclusion is not rigorous enough, and the conclusions (1), (2) and (3) are improved again, as follows:

(1) The thick-walled tube pneumatic fatigue impact loading precision separation process proposed in this paper effectively combines radial cyclic impact load, stress concentration effect and fatigue fracture principle, which realizes the thick-walled tube fatigue fracture.

(2) In the analysis of crack initiation at the bottom of V-notch in thick-walled tube, the threshold value of crack initiation at the tip of V-notch was analyzed when the tube was subjected to bending load F and bending moment M under plane strain state, and the expression of the corresponding theoretical stress concentration coefficient was obtained when the V-notch opening angle was . And the relationship between the impact load and the loading frequency of the tube and the bending moment. Based on this, it can be judged whether the fatigue crack at the root of the tube V-notch is cracked.

(3) Under the cyclic action of pneumatic impact load, the control method of gradually reducing the loading frequency from high to low can effectively control the initiation, stable expansion and rapid rupture of thick-walled tube cracks. Implements tube fatigue fracture, and combining with thick wall tube fatigue fracture microscopic morphology, fracture analysis of the thick wall tube in the section of pneumatic impact fatigue loading precision separation characteristics of the GCr15 seamless steel tube is obtained by measuring and data processing section figure 3D data, contrast analysis of the initial frequency and terminate the relationship with the fracture quality. According to the proposed fracture quality evaluation method, the fracture quality of the separation section of the thick-walled tube is effectively analyzed.

13、I think it is good to try a new separation method in this study.

However, as mentioned above, it is judged that there are many insufficiencies in the results of the study in general, and the discussion of the results is also insufficient.

I think it is necessary to further improve the completeness of the paper.

Authors’ response:

Thanks a lot for the reviewer’s comments.

The author has revised the "Conclusion" of this article to read:

(1) The thick-walled tube pneumatic fatigue impact loading precision separation process proposed in this paper effectively combines radial cyclic impact load, stress concentration effect and fatigue fracture principle, which realizes the thick-walled tube fatigue fracture.

(2) In the analysis of crack initiation at the bottom of V-notch in thick-walled tube, the threshold value of crack initiation at the tip of V-notch was analyzed when the tube was subjected to bending load F and bending moment M under plane strain state, and the expression of the corresponding theoretical stress concentration coefficient was obtained when the V-notch opening angle was . And the relationship between the impact load and the loading frequency of the tube and the bending moment. Based on this, it can be judged whether the fatigue crack at the root of the tube V-notch is cracked.

(3) Under the cyclic action of pneumatic impact load, the control method of gradually reducing the loading frequency from high to low can effectively control the initiation, stable expansion and rapid rupture of thick-walled tube cracks. Implements tube fatigue fracture, and combining with thick wall tube fatigue fracture microscopic morphology, fracture analysis of the thick wall tube in the section of pneumatic impact fatigue loading precision separation characteristics of the GCr15 seamless steel tube is obtained by measuring and data processing section figure 3D data, contrast analysis of the initial frequency and terminate the relationship with the fracture quality. According to the proposed fracture quality evaluation method, the fracture quality of the separation section of the thick-walled tube is effectively analyzed.

(4) The separation device of pneumatic fatigue impact loading precision separation process has a simple structure, is easy to operate, saves raw materials, low energy consumption and obtains a good thick-walled tube precision separation effect, which has certain engineering application significance.

Reviewer 2 Report

The manuscript is on topic in the field of thick wall pipe separation under the cyclic action of pneumatic impact load. Experimental results are presented that describe the features of loading and separation of a GCr15 steel pipe with an outer diameter of 30 mm and a wall thickness of 6 mm into segments of 50 mm in length. The research results are interesting, relevant and the manuscript is recommended for publication. However, several points should be clarified before publishing.

1. Load values F and load change intervals ΔF are not specified in the paper. The authors write (Lines 133-135): “Therefore, the regulation of the impact load size and loading frequency of the tube can control the initiation and expansion of cracks.” Loading frequency is analyzed in the paper and the main attention of the authors is devoted to this, but “impact load size” is not analyzed. Equations for F and ΔF (Equations (11) and (12)) are given, but the values of these parameters are not specified.

2. There is a problem with the denomination of the “the groove depth” parameter. Line 91, Equations (3), (5), page 4, Line 167, Table 1 - this parameter is denominated as “t”. Figure 2, Line 161, Figure 4 - this parameter is denominated as “h”.

3. Line 102. What does “R” mean?

4. There is a problem with Table 2. Obviously, the last two columns are unnecessary. There is no line with "Test number IV".

5. Table 2, Line 190. It is not clear how the termination frequency 4.5 Hz and 3.5 Hz were obtained if the initial frequencies were 15 and 25 Hz, respectively, and the frequency change interval was 2 Hz?

6. Line 291. It's not clear where the value of 13 Hz comes from?

7. Line 299. Obviously, instead of "Fig. 10(b)" there should be " Fig. 10(a)".

8. “Reference”. The authors do not make any references to papers from the journal to which they submit the manuscript. How can this be explained? Are the authors unfamiliar with the papers from this journal? The topic of the manuscript does not coincide with the topic of the journal?

Author Response

1.Load values F and load change intervals ΔF are not specified in the paper. The authors write (Lines 133-135): “Therefore, the regulation of the impact load size and loading frequency of the tube can control the initiation and expansion of cracks.” Loading frequency is analyzed in the paper and the main attention of the authors is devoted to this, but “impact load size” is not analyzed. Equations for F and ΔF (Equations (11) and (12)) are given, but the values of these parameters are not specified.

Authors’ response:

Thanks a lot for the reviewer’s comments.

The crack initiation load of fatigue crack on metal thick-walled tube is related to the fatigue fracture performance of the material itself and the geometric parameters of the V- notch of the tube, while the instantaneous fracture load of fatigue crack is only related to the fatigue fracture performance and the length of the fatigue crack. The load F required for crack initiation of the tube in this paper is 5000N, and the load interval :0~6000N.

2.There is a problem with the denomination of the “the groove depth” parameter. Line 91, Equations (3), (5), page 4, Line 167, Table 1 - this parameter is denominated as “t”. Figure 2, Line 161, Figure 4 - this parameter is denominated as “h”.

Authors’ response:

Thanks a lot for the reviewer’s comments.

The parameters corresponding to "the groove depth" have been unified in the text, and they are all expressed as "h".

3.Line 102. What does “R” mean?

Authors’ response:

Thanks a lot for the reviewer’s comments.

R in the text represents the stress ratio. In this study, the fatigue impact load is essentially a cyclic alternating load, the fatigue load in one cycle is called a stress cycle, the minimum stress value in a stress cycle is referred to as the minimum stress , and the maximum stress value is referred to as the maximum stress. Stress , the stress cycle can be described by some parameters, such as the stress ratio R. The ratio of the minimum stress to the maximum stress in the stress cycle is called the stress ratio R.

4.There is a problem with Table 2. Obviously, the last two columns are unnecessary. There is no line with "Test number IV".

Authors’ response:

Thanks a lot for the reviewer’s comments.

We apologize for this ignorance, it has been corrected, and we have checked and corrected all potential errors throughout the text.

Table 2. GCr15 steel tube separation experimental parameters

Test number

(Hz)

(Hz)

(Hz)

(s)

16.5

4.5

2

4

20

4

2

4

25.5

3.5

2

4

31

3

2

4

5.Table 2, Line 190. It is not clear how the termination frequency 4.5 Hz and 3.5 Hz were obtained if the initial frequencies were 15 and 25 Hz, respectively, and the frequency change interval was 2 Hz?

Authors’ response:

Thanks a lot for the reviewer’s comments.

The test data was re-verified, and the initial frequencies were 15, 25 Hz and 30 Hz in Test No. I in Table 2 were corrected to 16.5 Hz, 25.5 Hz and 31 Hz.

6.Line 291. It's not clear where the value of 13 Hz comes from?

Authors’ response:

Thanks a lot for the reviewer’s comments.

The initial frequency is the frequency at which the tube is initially loaded during the experiment, and the frequency duration refers to the time for continuous loading while keeping the frequency value unchanged at this frequency. As shown in the figure, is the frequency continuous loading time. Figure 10(b) shows the relationship between the stop frequency and the cross section. If there are multiple stop frequencies, there should be corresponding initial frequencies. The initial frequencies corresponding to all stop frequencies here are selected as 13Hz as the value of the experimental study, only It is only the initial frequency corresponding to the stop frequency, and has no special meaning.

7.Line 299. Obviously, instead of "Fig. 10(b)" there should be " Fig. 10(a)".

Authors’ response:

Thanks a lot for the reviewer’s comments.

We have made changes in the text from "Fig. 10(b)" in line 299 to "Fig. 10(a)".

8.“Reference”. The authors do not make any references to papers from the journal to which they submit the manuscript. How can this be explained? Are the authors unfamiliar with the papers from this journal? The topic of the manuscript does not coincide with the topic of the journal?

Authors’ response:

Thanks a lot for the reviewer’s comments.

References related to the research content in this journal have been added to the "Reference" as follows:

  1. Xiao, S.H.; Luan, X.S.; Liang, Z.Q.; Wang, X.B.; Zhou, T.F.; Ding, Y. Fracture Analysis of Ultrahigh-Strength Steel Based on Split Hopkinson Pressure Bar Test. Metals. 2022, 12(4), 628.
  2. Jukic, K.; Peric, M.; Tonkovic, Z.; Skozrit, I.; Jarak, T. Numerical Calculation of Stress Intensity Factors for Semi-Elliptical Surface Cracks in Buried-Arc Welded Thick Plates. Metals. 2021, 11(11), 1809.
  3. Yang, M.; Gao, C.; Pang, J.; Li, S.; Hu, D.; Li, X.; Zhang, Z. High-Cycle Fatigue Behavior and Fatigue Strength Prediction of Differently Heat-Treated 35CrMo Steels. Metals. 2022, 12,688.

Reviewer 3 Report

In this paper, the authors proposed a new method for thick-walled metal tube precision separation using pneumatic fatigue impact loading. This was proposed keeping in mind the limitations of the traditional separation methods (turning, sawing), i.e., low efficiency and substantial raw material wastage.

The scientific scope of the study is not clear as there are other advanced cutting methods (e.g., wire EDM, laser cutting) which have significantly less wastage of material. The method proposed by the author has many loopholes (briefly mentioned below). The manuscript does not resemble a research paper. The structure is not organized well (introduction, materials and methods, results, discussions, conclusions), and many grammatical errors are there in the manuscript. Details about the findings are also missing. Therefore, I am resigned to recommend a rejection instead of a transfer.

A few comments on some technical aspects the authors should focus on if they wish to convert this manuscript to a paper:

  1. The main goal of this manuscript is to propose a tube precision separation method with minimal raw material wastage. However, I am unable to understand how manufacturing of V-groove for pneumatic fatigue impact loading saves the material.
  2. Another issue with this new method of tube separation (pneumatic fatigue impact loading) is the surface finish. The surface finish of the tube sections obtained from the tests, shown in Fig. 6, appears to be poor. The dimensions of S1 (convex area) and S2 (concave area) (shown in Fig. 8) are in order of mm (Table 3), which also indicates that the method is not much precise. Some advanced cutting methods (e.g., wire EDM, laser cutting, waterjet cutting) give a better surface finish with less raw material wastage. It is not clear how the pneumatic fatigue impact loading method for tube separation is a near-net-shape separation method.
  3. There is an additional factor that needs to be critically assessed while striking the tube. Continuous strikes of impact hammers can alter the stress state of the material and make the tube more susceptible to failure. Which application needs this kind of material? Is this pneumatic fatigue impact loading applicable to cutting all kinds of materials?
  4. The length of the tube can increase when the impact hammer is stricken in order to conserve the volume. The authors must consider this aspect and suggest a way by which it can be prevented.
  5. The hardness of the surface increases because of the compressive residual stresses. The authors should measure hardness before and after the test through indentation or other techniques.
  6. The V-groove area is minimal (ρ=0.2 mm, t=1.2 mm) (t is not defined in the manuscript), while the hammer size is big (180mm diameter). Load is not fully concentrated on the groove. What is the rationale for selecting bigger sized hammers?
  7. How is the machine calibrated?
  8. What is the rationale for selecting six impact hammers? How much load is required to complete the separation? How much time is required to produce the two parts of the tube? How many cycles of loading are required to separate the tube? Is the load constant throughout the process or varying continuously?
  9. Which type of fracture is occurring? More explanation about fractography is needed.
  10. Huge noise will be created. How to control it?
  11. On page 7, Table 2 is not having details for test number IV.
  12. On page 3, line 84: When △KI < △Kth, the crack does not crack. Crack does not crack?
  13. On page 2, line 62: First of all, the lathe and other processing machines turn the annular V-notch with specific geometric parameters on the specific position of the tube raw material. Which method is used by authors to create v-notch?
  14. Page 9, line 227: a1% and a2% are their percentages of the entire tube section area. The statement is vaguely written.

Author Response

1.The main goal of this manuscript is to propose a tube precision separation method with minimal raw material wastage. However, I am unable to understand how manufacturing of V-groove for pneumatic fatigue impact loading saves the material.

Authors’ response:

Thanks a lot for the reviewer’s comments.

The research object of this precise separation method is the thick-walled tube, and the V-groove only hopes to increase a stress concentration effect on the outer surface of the tube, and the material removed is relatively small. The ultimate goal is to make precise separation of tubes by studying fatigue fracture.

2.Another issue with this new method of tube separation (pneumatic fatigue impact loading) is the surface finish. The surface finish of the tube sections obtained from the tests, shown in Fig. 6, appears to be poor. The dimensions of S1 (convex area) and S2 (concave area) (shown in Fig. 8) are in order of mm (Table 3), which also indicates that the method is not much precise. Some advanced cutting methods (e.g., wire EDM, laser cutting, waterjet cutting) give a better surface finish with less raw material wastage. It is not clear how the pneumatic fatigue impact loading method for tube separation is a near-net-shape separation method.

Authors’ response:

Thanks a lot for the reviewer’s comments.

The surface finish of the tube section obtained by the aerodynamic fatigue impact loading precision separation method proposed in this paper is already better than other methods currently in the research stage, such as the rotary wedge shearing method and the impact shearing method (as shown in Figure 11 in the paper). As for the wire EDM, laser cutting, and waterjet cutting mentioned by the reviewers, they are not considered advanced cutting methods and have been industrialized for a long time. Furthermore, these methods are either too energy-intensive or expensive, and are not suitable for large-volume tubes. Cutting, which is also the main reason why manufacturers who have a large number of tube cuttings do not use it. The background of the tube precision separation method proposed in this paper is green manufacturing and low-carbon processing, which are the requirements and needs of industrialization development. As for the surface finish of the tube, in many applications, the finish of the end face of the tube is not high, such as the transmission chain, the sleeve in the conveyor belt, etc., the finish of the end face is relatively low.

3.There is an additional factor that needs to be critically assessed while striking the tube. Continuous strikes of impact hammers can alter the stress state of the material and make the tube more susceptible to failure. Which application needs this kind of material? Is this pneumatic fatigue impact loading applicable to cutting all kinds of materials?

Authors’ response:

Thanks a lot for the reviewer’s comments.

The precision separation method of aerodynamic fatigue impact loading proposed in this paper requires the impact of the hammer head when radially loading the tube. After a lot of experiments, the hammer head did not cause obvious damage to the surface of the tube during the impact process, but this process still as research continues, adding a protective sheath to the outer surface of the tube will be considered in subsequent studies. Most of the bearing rings and rolling elements are made of GCr15 material. This research is aimed at the blanking method of the sleeve and bearing ring of the transmission chain. The research is still in the process of in-depth progress, and the follow-up will also try other types of pipes and tubes, and carry out research on the precise separation of tubes of various materials.

4.The length of the tube can increase when the impact hammer is stricken in order to conserve the volume. The authors must consider this aspect and suggest a way by which it can be prevented.

Authors’ response:

Thanks a lot for the reviewer’s comments.

The object of this paper is the GCr15 thick-wall tube. First of all, the tube wall is thick and the material hardness is high. When the tube is subjected to radial impact loading, a sleeve is added to its outer surface for protection, with the main purpose of preventing the impact load from producing obvious plastic deformation on the outer surface of the tube. No obvious plastic deformation was found in the experimental process, so there is no need to worry about the plastic deformation of the GCr15 thick wall tube when it is separated.

5.The hardness of the surface increases because of the compressive residual stresses. The authors should measure hardness before and after the test through indentation or other techniques.

Authors’ response:

Thanks a lot for the reviewer’s comments.

Due to the precision separation method of aerodynamic fatigue impact loading proposed in this paper, the compressive stress on the surface of the tube will change after multiple hammer blows on the outer surface of the tube, and the test of residual compressive stress will be added later, but the change of the surface stress of the tube is not the original research considerations

6.The V-groove area is minimal (ρ=0.2 mm, t=1.2 mm) (t is not defined in the manuscript), while the hammer size is big (180mm diameter). Load is not fully concentrated on the groove. What is the rationale for selecting bigger sized hammers?

Authors’ response:

Thanks a lot for the reviewer’s comments.

Here t=1.2mm refers to the groove depth of the tube. The author has made a unified mark on the groove depth in the article, and uses h as the unified symbol for the groove depth of the V-shaped groove. There are two main reasons for choosing a relatively large-sized hammer in this study: (1) Since the research object is the precise separation of thick-walled tubes, it is necessary to apply a large impact load to the tube to promote its rapid initiation of cracks and improve the Precision separation efficiency; (2) In the third question above, the review experts have raised questions about whether the outer surface of the tube is damaged. The author chooses the largest possible size of the hammer just to minimize the damage to the outer surface of the tube. Hammer loading, too small a size will cause significant damage to the outer surface of the tube.

Although the load is not completely concentrated on the groove, in this process, the tube is a cantilever beam structure, and there is a corresponding specially designed fixing fixture. The clamping position is also particular. Through research, it is found that the fixture must be fixed after the groove. At the edge, the force will work better, and most of the load will be applied to the groove of the tube.

7.How is the machine calibrated?

Authors’ response:

Thanks a lot for the reviewer’s comments.

The fixed position of the tube clamp is one of the important benchmarks for calibration.

8.What is the rationale for selecting six impact hammers? How much load is required to complete the separation? How much time is required to produce the two parts of the tube? How many cycles of loading are required to separate the tube? Is the load constant throughout the process or varying continuously?

Authors’ response:

Thanks a lot for the reviewer’s comments.

(1) In the study of tube precision separation in this paper, the pneumatic fatigue impact loading tube precision separation device is mainly composed of six double-acting single-piston striking cylinders evenly distributed in the circumference of the cylinder block and six striking hammers connected with the cylinder piston. In the design of the experimental device, not only the radial loading of the tube can be considered, but also the cyclic action of the force from different angles can be realized, that is to say, different hammers can be controlled for radial loading, and the loading sequence of the hammers can also be adjusted. After comprehensive consideration, 6 hammers are reasonable. If the number of hammers is too small, it will affect the research and control of different loading sequence of hammers. If the number of hammers is too large, the structure will be too complicated and the control will be too complicated.

(2) The initiation load of fatigue crack on metal thick-walled tube is related to the fatigue fracture property of the material itself and the geometric parameters of the v-notch of the tube, while the transient load of fatigue crack is related to the fatigue fracture property and the length of the fatigue crack. The load required for tube cracking in this paper is 5000N.

(3) The time required for tube separation is related to experimental parameters. The time required for tube separation in this paper is shown in Table 3, which are 45s, 33s, 35s and 20s respectively.

Table 3. Section parameters and separation time of GCr15 steel tube

Test number

h1(mm)

h2(mm)

a1(%)

a2(%)

tc(s)

0.76

0.16

13.6

0.15

45

0.68

0.13

12.5

0.12

33

0.22

0.15

0.21

0.09

35

0.17

0.12

0.20

0.08

20

(4) separation tube loading cycles and the geometric parameters of tube v-notch, the initial frequency and the termination of the frequency-dependent, and different loading sequence, can also lead to loading cycle is different, each factor changes will lead to loading cycles is different, so in the process of the research focuses on tubing separation time required.

(5) In this study, the load can be kept constant or continuously changed during the whole loading process. In short, the load can be adjusted during the whole loading process. In this paper, "1 Principle of pneumatic impact loading separation" also describes the characteristics of the load in the loading process, "each impact hammer is driven by an independent cylinder and piston, by changing the pressure and flow of the fluid in the cylinder output different load size and speed of the impact force".

9.Which type of fracture is occurring? More explanation about fractography is needed.

Authors’ response:

Thanks a lot for the reviewer’s comments.

So far, through a large number of experimental studies, only one kind of fracture mode of fatigue fracture has occurred in the tube. In the follow-up, we will continue to pay attention to the fracture method of the tube and the fracture characteristics of the tube.

10.Huge noise will be created. How to control it?

Authors’ response:

Thanks a lot for the reviewer’s comments.

The precision separation method of aerodynamic fatigue impact loading proposed in this paper will indeed generate noise during the test. The research is still in the early stage and is carried out in an experimental environment far away from the office area and the teaching area. There is no better way to eliminate the noise. In the follow-up research, we will seriously consider this problem, and strive to find a more suitable noise reduction method.

11.On page 7, Table 2 is not having details for test number IV.

Authors’ response:

Thanks a lot for the reviewer’s comments.

We have supplemented Table 2 by adding "Test number IV".

Table 2. GCr15 steel tube separation experimental parameters

Test number

(Hz)

(Hz)

(Hz)

(s)

16.5

4.5

2

4

20

4

2

4

25.5

3.5

2

4

31

3

2

4

12.On page 3, line 84: When △K< △Kth, the crack does not crack. Crack does not crack?

Authors’ response:

Thanks a lot for the reviewer’s comments.

Whether the bottom of the V-notch on the thick-walled tube cracks is related to the size of the stress intensity factor ( ) at the crack tip and the size of the threshold value of crack propagation ( ). When the stress intensity factor is less than the threshold value of crack propagation, the bottom of the V-notch on the thick-walled tube does not crack.

13.On page 2, line 62: First of all, the lathe and other processing machines turn the annular V-notch with specific geometric parameters on the specific position of the tube raw material. Which method is used by authors to create v-notch?

Authors’ response:

Thanks a lot for the reviewer’s comments.

In the current study, V-notch was turned by a lathe due to limited experimental conditions. However, during the research process, the team members have already paid attention to the machining of annular V-shaped notches. This method still needs to be optimized, and we plan to explore a new method of prefabricated stress concentration notches in the follow-up research.

14.Page 9, line 227: a1% and a2% are their percentages of the entire tube section area. The statement is vaguely written.

Authors’ response:

Thanks a lot for the reviewer’s comments.

It has been more clearly stated in the paper and modified as: a1% and a2% are the percentages of the convex area (S1) and the concave area (S2) of the entire tube area, which can be calculated based on the measurement data of the section planeness.

Round 2

Reviewer 1 Report

[1] from the author's answer

1) It can be seen from Equation 12 that the bending moment of the tube during the impact loading process is related to the mass of the impact hammer and the frequency of the impact load. However, only the frequency of the impact load is a variable, so this paper focuses on the pipe material. In the process of separation experiment research, mainly starting from adjusting the frequency of impact load, many tests and analyses were carried out.

2) If it is necessary to analyze whether the fatigue crack starts or not, it can be analyzed with the formula derived theoretically in this paper. The specific judgment method is to substitute formulas (9) and (12) into formula (5) and compare with (1), then determine whether the fatigue crack at the root of the V-notch of the tube is cracked.

It would be better to indicate above mentions (1) and (2) as the text content in paper.

[2] from the author's answer

And, it would be good to show the graph in the text and mention the contents that presented in the author's answer follows as "(3) The relationship between the magnitude of the impact load and the crack initiation in this paper is shown in the figure.".

[3] from the author's answer

The initial frequency is the frequency at which the tube is initially loaded during the experiment, and the frequency duration refers to the time for continuous loading while keeping the frequency value unchanged at this frequency. As shown in the figure, Dt is the frequency continuous loading time. And the typo in the text "frequency continuous loading time s Dt " has been changed to "frequency continuous loading time Dt ". Figure 10(b) shows the relationship between the stop frequency and the cross section. If there are multiple stop frequencies, there should be corresponding initial frequencies. The initial frequencies corresponding to all stop frequencies here are selected as 13Hz as the value of the experimental study, only It is only the initial frequency corresponding to the stop frequency, and has no special meaning.

Above mentions it would be good if the contents from the author's answer were corrected in the main text.

Author Response

Reviewer #1:

[1] from the author's answer

1) It can be seen from Equation 12 that the bending moment of the tube during the impact loading process is related to the mass of the impact hammer and the frequency of the impact load. However, only the frequency of the impact load is a variable, so this paper focuses on the pipe material. In the process of separation experiment research, mainly starting from adjusting the frequency of impact load, many tests and analyses were carried out.

2) If it is necessary to analyze whether the fatigue crack starts or not, it can be analyzed with the formula derived theoretically in this paper. The specific judgment method is to substitute formulas (9) and (12) into formula (5) and compare with (1), then determine whether the fatigue crack at the root of the V-notch of the tube is cracked.

It would be better to indicate above mentions (1) and (2) as the text content in paper.

Authors’ response:

Thanks a lot for the reviewer’s comments.

We have added (1) and (2) to the manuscript.

[2] from the author's answer

And, it would be good to show the graph in the text and mention the contents that presented in the author's answer follows as "(3) The relationship between the magnitude of the impact load and the crack initiation in this paper is shown in the figure.".

Authors’ response:

Thanks a lot for the reviewer’s comments.

We have added content to the manuscript.

[3] from the author's answer

The initial frequency is the frequency at which the tube is initially loaded during the experiment, and the frequency duration refers to the time for continuous loading while keeping the frequency value unchanged at this frequency. As shown in the figure, Dt is the frequency continuous loading time. And the typo in the text "frequency continuous loading time s Dt " has been changed to "frequency continuous loading time Dt ". Figure 10(b) shows the relationship between the stop frequency and the cross section. If there are multiple stop frequencies, there should be corresponding initial frequencies. The initial frequencies corresponding to all stop frequencies here are selected as 13Hz as the value of the experimental study, only It is only the initial frequency corresponding to the stop frequency, and has no special meaning.

Authors’ response:

Thanks a lot for the reviewer’s comments.

We have added content to the manuscript.

Reviewer 3 Report

The paper has been modified satisfactorily

Author Response

Dear editors,

On behalf of my co-authors, we would like to give the author’s great thanks to you and the anonymous reviewers for the valuable suggestions, which have been carefully taken into account during the revision of our manuscript metals-1692413. Please find enclosed a description of the changes made to this paper, which is in response to the reviewers’ comments.

We would like to express our great appreciation to you and the reviewers for comments on our paper, and we are looking forward to hearing from you.